# Tissue-Engineered Models for Glaucoma Research

**DOI:** 10.3390/mi11060612

**Published:** 2020-06-24

**Authors:** Renhao Lu, Paul A. Soden, Esak Lee

**Affiliations:** 1Nancy E. and Peter C. Meinig School of Biomedical Engineering, Cornell University, Ithaca, NY 14853, USA; rl839@cornell.edu; 2College of Human Ecology, Cornell University, Ithaca, NY 14853, USA; pas325@cornell.edu

**Keywords:** glaucoma, tissue engineering, trabecular meshwork, Schlemm’s canal, retinal ganglion cell, intraocular pressure, optic nerve head, electrospinning, soft lithography, 3D scaffold, 3D bioprinting

## Abstract

Glaucoma is a group of optic neuropathies characterized by the progressive degeneration of retinal ganglion cells (RGCs). Patients with glaucoma generally experience elevations in intraocular pressure (IOP), followed by RGC death, peripheral vision loss and eventually blindness. However, despite the substantial economic and health-related impact of glaucoma-related morbidity worldwide, the surgical and pharmacological management of glaucoma is still limited to maintaining IOP within a normal range. This is in large part because the underlying molecular and biophysical mechanisms by which glaucomatous changes occur are still unclear. In the present review article, we describe current tissue-engineered models of the intraocular space that aim to advance the state of glaucoma research. Specifically, we critically evaluate and compare both 2D and 3D-culture models of the trabecular meshwork and nerve fiber layer, both of which are key players in glaucoma pathophysiology. Finally, we point out the need for novel organ-on-a-chip models of glaucoma that functionally integrate currently available 3D models of the retina and the trabecular outflow pathway.

## 1. Introduction

Glaucomas are a heterogeneous group of optic neuropathies characterized by the progressive degeneration of retinal ganglion cells (RGCs) [1]. The gradual decline in retinal integrity associated with glaucoma can leave affected individuals with a spectrum of visual deficits, and glaucoma now represents the leading cause of irreversible blindness worldwide. It is estimated that more than 70 million people will be affected by glaucoma by 2020, with approximately 10% of cases progressing to total bilateral sight loss [2]. Moreover, the treatment of glaucoma in the United States (US) alone incurs an economic cost of over $1.5 billion dollars annually, most of which is spent on outpatient ophthalmic care to monitor the disease’s progression [3].

The two main types of glaucoma—open-angle and angle-closure glaucoma—are distinguished based upon the optical structures which they affect. Primary open-angle glaucoma (POAG) accounts for most cases in Western Europe, Africa and the US [4], while primary angle-closure glaucoma (PACG) is more common among Asians [5]. However, both types of glaucoma are related to a core cluster of common risk factors, including age, race, family history of glaucoma, myopia and elevated intraocular pressure (IOP) [6]. Among these, IOP elevation serves as the most common risk factor for glaucoma and is the primary target of pharmacological and surgical intervention [7].

IOP is thought to elevate primarily as a consequence of changes in the resistance of the Trabecular meshwork (TM) and the inner wall endothelium of Schlemm’s canal (SC), both of which normally drain aqueous humor (AH) fluids from the intraocular space [8]. However, despite years of investigation on this subject, the precise molecular and biophysical mechanisms by which these pathologic changes contribute to IOP elevation remain unclear. This is due in part to the limitations associated with using conventional 2D cell culture and anterior segment perfusion models to investigate the relative contributions of biologic and biophysical factors to AH outflow. A better mechanistic understanding of elevated IOP would facilitate the development of novel preventative approaches for glaucoma and allow clinicians to identify at-risk patients even earlier in the disease process.

Similarly, current in vitro models of retinal and optic nerve head degeneration in glaucoma are limited in their physiological relevance. This has thus far hindered efforts to understand, treat and prevent the pathologic changes that underlie glaucoma-related anopsia. Furthermore, it has been particularly difficult to investigate how the mechanisms of RGC degeneration differ between patients with and without elevated IOP [9]. While RGC loss in glaucoma is believed to be irreversible, implantation of cell grafts under the appropriate conditions have shown early success in preclinical models at somewhat restoring retinal integrity [10]. A better understanding of the molecular details associated with RGC damage and how these details differ between individual glaucoma patients, will inform the development of more targeted, patient-specific cell therapies. This all highlights the need for tissue engineered models of glaucomatous physiology that more closely recapitulate the AH outflow pathways and RGC degeneration observed in vivo.

In this review, we begin with a brief overview of basic eye physiology and glaucoma pathogenesis. We then compare current tissue engineered models of AH outflow and RGC degeneration, specifically highlighting those which mobilize soft lithography, electrospinning, microfluidics, hydrogel 3D scaffolds and 3D bioprinting technologies. Integrating this information, we propose a novel, three-dimensional (3D) organ-on-a-chip model of glaucomatous physiology that combines currently available models of both trabecular outflow and RGC degeneration. We argue that this platform will help us better understand how IOP contributes to RGC degeneration during glaucoma progression. Finally, we also describe how our proposed model better recapitulates the in vivo cell–cell and cell-extracellular matrix (ECM) interactions that mediate glaucoma-related vision loss.

## 2. Basic Physiology of Eye

In this section, we briefly introduce the basic principles of ocular physiology that are relevant to glaucoma pathogenesis. As is shown in Figure 1, the intraocular space (~6000 μL) is an immune-privileged environment that can be subdivided into three distinct sections: the anterior chamber (between the cornea and the iris), the posterior chamber (between the iris and lens) and the vitreous chamber (between lens and the back of the eye) [11]. The anterior chamber (~250 μL) and posterior chamber (~60 μL) are filled with the AH, a transparent, non-viscous fluid [12]. By contract, the vitreous chamber is occupied by the vitreous humor, a non-homogenous, viscoelastic gel made up primarily of hydrated ECM fibrils [13]. Both the aqueous and vitreous humor are impressible materials contained within the optic globe by concentrically arranged membranes. These structures are flanked externally by the sclera (five sixths of globe), and the cornea (one sixth of the globe), and the elastic properties of these two membranes confer the eye with a sizeable rigidity. In fact, the mean ocular rigidity coefficient is estimated to be around 0.77 mmHg/μL, meaning that IOP is highly sensitive to even small shifts of intraocular volume [11]. Lining the back of the eye is the retina, which contains the light-sensitive photoreceptors that transduce visual stimuli into neuronal signals. The mammalian retina can be thought of as an “inverted retina” because its photoreceptors (rods and cones) are located *behind* the other neuronal cells involved in the neurotransmission of visual data. Namely, visual data are sent from the photoreceptors through bipolar, amacrine and horizontal cells to eventually reach the RGCs, the axons of which make up the afferent optic nerve [14].

### 2.1. Fluid Dynamics of AH

As mentioned earlier, raised IOP is a known precursor of glaucoma thought to result from perturbations in the fluid dynamics of the AH. Under non-pathologic conditions, the proper turnover of AH fluids plays a critical role in supporting the shape of the optic globe, maintaining a healthy IOP and promoting the refractory properties of the eye [12,15]. Moreover, AH circulation removes wastes from and supplies oxygen, nutrients, and neurotransmitters to the avascular tissues of the anterior eye, including the cornea and lens. In addition to its high nutrient density, AH fluids are rich in ascorbic acid, but very low in proteins. Its soluble proteome is known to consist primarily of plasma proteins, transthyretin, ceruloplasmin, proteases, protease inhibitors, neuropeptides, anti-angiogenic proteins, chondromodulin and steroid-converting enzymes [11,16].

After being secreted from the ciliary epithelium in the ciliary body, AH flows around the lens and through the iris, eventually draining into the anterior chamber angle via the conventional or unconventional outflow pathway [17]. In human eyes, the rate of AH turnover is subject to a circadian rhythm. In fact, morning flow rates are known to be almost double those of nighttime, possibly mediated by cyclical increases in basal epinephrine levels during the day [12,18]. Additionally, the rate of AH secretion is known to decline by about 2.4% per decade between the ages of 20 and 80 years old [19,20]. However, IOP should remain within the normal range for most of the lifespan, as a concurrent increase in outflow resistance also occurs with age [21]. The 0.25 mL of fluid that make up a healthy AH therefore maintains a fairly stable turnover rate of 2.4 μL/min [22].

#### 2.1.1. AH Production

AH fluids are secreted by the ciliary body, a muscular structure that circumscribes the iris. The primary secretion surface of the ciliary body is the ciliary process, which contains an external, pigmented epithelium and an internal, non-pigmented epithelium [12,23]. The non-pigmented epithelium is knitted together by tight junctions, forming a selective exchange surface across which ions and water from the ciliary capillary bed are secreted into the anterior chamber. During the production of AH, ions are pumped across this surface through several active transport processes [11]. Most notably, the sodium–potassium ATPase and carbonic anhydrase critically mediate the transport of cations and bicarbonate, respectively, across the non-pigmented epithelium [12,24]. Such transport processes set up an electrostatic gradient down which chloride anion can flow into the intraocular space [24].

Furthermore, the buildup of solutes in the intraocular space draws water down its osmotic gradient into the anterior and posterior chambers through aquaporin channels [12,25]. Selective transport of glutathione and nutrition compounds also occurs across the non-pigmented epithelium [26]. The limited protein complement of the AH is furnished by a pressure-dependent “ultrafiltration” of blood through the fenestrated endothelium of the ciliary capillaries [12,27], a process which decreases as IOP rises [28]. Finally, diffusion processes may mediate the transport of some lipophilic compounds from the ciliary body into the intraocular space [12,29]. The ciliary body is also believed to be an important source of paracrine signals that regulate the rate of AH turnover [11].

#### 2.1.2. AH Outflow

The efflux of AH fluids from the intraocular space primarily occurs through two anatomically distinct pathways. Although there is AH fluid and ion exchange in cornea, iris and vitreoretinal interface, no significant net fluid movement is found. In the trabecular pathway, also known as the direct or conventional outflow pathway, AH flows through the multilayered TM and the inner wall of SC into the lumen of SC, where it will pass through collector channels and reenter systemic circulation via the episcleral venous plexus and the aqueous veins of Ascher [30] (Figure 2). AH in the uveoscleral pathway, also known as the indirect or unconventional outflow pathway, is taken up by the iris root and flows through the uveal meshwork, the anterior face of ciliary muscle and the connective tissues between the muscle bundles. The fluid will eventually pass through the supra choroidal space and the sclera to be taken up by the ocular vasculature [8,31].

Generally, the trabecular pathway is considered to be pressure-dependent and drains 90% of the AH fluids. Moreover, the TM and inner wall of SC provide the resistance to AH outflow required to generate a baseline intraocular pressure. As mentioned earlier, the elevated IOP found in open angle glaucoma can be attributed to pathologic changes in the resistance properties of these two tissues [32]. Beyond its functions in IOP maintenance, the TM secretes ECM components that critically maintain the structural integrity of the eye. It also contains phagocytes which clear debris from the intraocular space [32]. SC is encased in a layer of elongated, spindle-shaped endothelial cells, creating the only continuous cell monolayer in trabecular pathway. Arranged along the external wall of the SC are 25–35 collector channels, which act as a conduit for AH flux into the episcleral venous plexus [33]. Interestingly, the flow of humoral fluids from the episcleral venous plexus into the aqueous vein is pulsatile, a functional consequence of blinking reflexes and the cardiac cycle [34].

In contrast to the trabecular pathway, the uveoscleral pathway is pressure-independent and drains only 10% of the AH fluids [8]. Currently, our understanding of the uveoscleral pathway and its physiological role is very limited. AH outflow via this pathway is believed to depend more on cyclic variations in the tone of ciliary muscles than on fluctuations in IOP [35]. In fact, relaxation of ciliary muscle has been shown to decrease IOP by enhancing uveoscleral outflow and several glaucoma drugs have been developed with this therapeutic mechanism in mind, including prostaglandin F_2α_ [36].

#### 2.1.3. AH Dynamics

The fluid dynamics that underlie IOP and AH outflow can be simplified through the use of mathematical models. Under non-pathologic conditions, AH inflow should equal AH outflow. In other words:(1)$$F_{in}=F_{out}=F_{uv}+F_{trab}$$

In Equation (1), *F_in_* and *F_out_* are the inflow and outflow rate of AH, respectively and *F_uv_* and *F_trab_* are the outflow rates from the uveoscleral and trabecular pathways, respectively. Uveoscleral outflow is assumed to be independent of the pressure forces that act on the intraocular space [37]. Conversely, trabecular outflow is dependent upon the resistance of the TM and the pressure difference between the intraocular space (the IOP) and the episcleral vein (*P_ev_*). Given its age and circadian-dependence, the resistance of TM tissues is known to give rise to a wide range of IOP values, which are accommodated by compensatory variations in AH outflow [38]. The IOP generated by the interplay between AH production by the ciliary body and AH efflux through the TM can be calculated as:(2)$$IOP=\left(F_{in}-F_{uv}\right)\cdot R_{trab}+P_{ev}$$

From Equation (2), it is obvious that the IOP measured at any given time is a reflection of the complex interplay between multiple external factors, including TM resistance, uveoscleral outflow, AH production and even variations in blood pressure [39]. Moreover, each of these factors represents viable targets for surgical or pharmacological intervention aiming to reduce IOP in the glaucomatous eye.

Considering the rigidity (k) of the intraocular space, IOP is also proportional to the volume of AH in the eye at any given time. So:(3)$$IOP=kV=k\int_{0}^{t}\left[F_{in}\left(\tau \right)-F_{uv}\left(\tau \right)-F_{trab}\left(\tau \right)\right]d\tau +kV_{0}$$

Combining Equations (2) and (3), we can derive a differential equation that describes the temporally dynamic properties of IOP. To simplify, *F_in_*, *F_uv_* and *P_ev_* are held constant:(4)$$\frac{d}{dt}IOP+\frac{k}{R_{trab}}IOP=k\left(F_{in}-F_{uv}+\frac{P_{ev}}{R_{trab}}\right)$$

While this model is highly simplified, it demonstrates that IOP is a fairly stable first order system—the balance point of which is supplied by the solution to Equation (2). This means that any fluctuations in IOP mediated by external forces will be automatically recovered back to the normal range by compensatory mechanisms. Our model also clearly conveys the key role that trabecular resistance plays in the regulation of IOP.

### 2.2. Retinal Ganglion Cells

RGCs are a specialized class of projection neurons that transmit visual information from retinal photoreceptors to the lateral geniculate nucleus of the brain. Their cell bodies cluster in the retinal ganglion cell layer (RGCL), a heterogeneous structure that varies in thickness between the fovea and the macula [40,41]. As is shown in Figure 3, RGCs synapse directly with bipolar and amacrine cells in the inner plexiform layer (IPL) within the retina, forming a peripheral integration center in which the receptive fields of several photoreceptors are consolidated [42]. Unmyelinated axons of the RGCs then cluster in parallel internally to form the nerve fiber layer (NFL), which travels laterally along the retinal surface towards the optic nerve head (ONH), also known as the optic disk.

The ONH provides a conduit through which NFL tissue can escape from the optic globe and travel to the brain. Once inside the ONH, ganglion cell axons aggregate to form a torus-shaped structure called the neuro retinal rim (NRR). This tissue travels perpendicular to the plane of the retina and eventually invades the lamina cribrosa—a matrix of connective tissue—to feed directly into the myelinated optic nerve [14]. Posterior to the lamina cribrosa, optic nerve fibers are surrounded by a subarachnoid space continuous with that of the central nervous system. This region, called the retrobulbar space, is kept under the constitutive influence of intracranial pressure (ICP), which itself is maintained by the hydrodynamics of cerebrospinal fluid (CSF) [43]. Furthermore, the hollow center of the NNR forms a cavernous indentation in the center of the ONH termed the optic cup. This structure generally appears as a white spot in the center of the optic disk when imaged [44].

Notably, the axonal processes of RGCs are known to represent almost 40% of the total cranial fibers that travel to the brain [45]. This, coupled with the high rate of neuronal transmission that occurs along these fibers, confers RGCs with unusually high metabolic demands [46]. Paradoxically, the requirement for optical transparency in front of the retina also severely restricts the total size and scope of the vasculature that serves this tissue. Such conflicting physiological requirements necessitates that a precise auto-regulatory system intrinsic to the eye maintains the ocular blood supply at all times [11].

In fact, the pressure and composition of the retinal bloodstream is tightly regulated by myogenic responses, [47] light-induced changes in blood flow [48] and several other vasoregulatory mechanisms [49]. Despite these compensatory measures; the high metabolic demands, lack of myelination, thin axonal diameter [50] and limited vasculature supply of the RGCL makes this tissue particularly sensitive to changes in intraocular pressure. Furthermore, the anatomic positioning of the RGCL as the innermost layer of the retina compounds its vulnerability to the aversive effects of raised IOP. Interestingly, rather than being sensitive to absolute IOP, ganglion cell axons seem to be particularly affected by the existence of pressure gradients between the intraocular space, the subarachnoid compartment [51] and the ocular blood supply [52,53,54,55].

## 3. Glaucoma Pathophysiology

Glaucoma is identifiable through a number of distinct clinical features. As the retinal ganglion cells of a glaucomatous eye die off, the RGCL ensheathing the retina begins to thin, a degenerative process which can be clinically measured using standard automated perimetry [56]. Furthermore, RGC degeneration also leads to atrophy of the NRR tissue. As the axonal fibers of this tissue decay and retreat outwards towards the disk margin, the optic cup is able to expand radially in a process known as optic nerve cupping [57]. The cup-to-disk ratio can be monitored using an ophthalmoscope, and values above 0.6–0.7 indicate potential glaucomatous changes [58]. Moreover, the expanding optic cup will also adopt an excavated appearance as connective tissues within the lamina cribrosa distort and bow posteriorly towards the optic nerve [26]. These morphologic changes in ocular physiology are often accompanied by hemorrhage of the disk rim [59], followed by peripheral vision loss and blindness. Interestingly, there is a temporal asynchrony between the neural tissue atrophy and vision loss associated with glaucoma. In fact, 30%–50% of the optic nerve can degenerate before any visual impairments are detected [26].

### 3.1. The Impact of Intraocular Pressure

Although multiple genetic and environmental factors are involved in glaucoma pathogenesis, elevated IOP is the most common pathway by which neuronal damage is initiated. Consequently, it is necessary to discuss the pressure environment of the intraocular space and its impact on the optic nerve tissues in further detail.

The ONH can be subdivided into three distinct sections: the prelaminar region, the laminar region (lamina cribrosa) and the retrolaminar region. Each of these structures protrudes through a posterior fenestration in the sclera and supports the nerve fibers and vascular tissues entering and exiting the eyeball. As mentioned earlier, the lamina cribrosa is a matrix of connective tissue in which dense collagen and elastin fibrils surround and support the RGC axons of the neuro retinal rim. Glial cells that also cluster in this region help separate efferent axons from retinal blood vessels, and in a healthy, non-glaucomatous eye, the lamina cribrosa should be continuous with the sclera [60]. Organelles, most notably mitochondria, also tend to accumulate in the axons of retinal ganglion cells where they cross the laminal cribrosa [61]. The prelaminar region has a similar architecture and function as the lamina cribrosa, although it is localized anterior to the scleral fenestration [62,63]. Finally, the retrolaminar region, which is posterior to the lamina cribrosa, contains oligodendrocytes that wrap around optic nerve fibers and form the myelin sheath [64].

As noted previously, retinal ganglion cells are sensitive to the forces generated by pressure gradients, but not to the absolute pressure of the intraocular space. However, the pressure within the prelaminar and retrolaminar regions remains fairly constant over time, and the only pressure gradient that has the potential to meaningfully contribute to glaucoma pathogenesis is that observed across the lamina cribrosa. The so-called trans-lamina cribrosa pressure (TLCP) of healthy individuals tends to fluctuate between 2–4 mmHg. However, this value often rises to much as 6–10 mmHg in the pre-glaucomatous and glaucomatous eye [65,66]. Such changes could be induced by increases in IOP, decreases in ICP or some combination of the two. Regardless, a shift in the TLCP that favors fluid-flow into the retrobulbar space can critically remodel the extracellular matrix, neuronal tissue and vasculature of the optic disk, leading to glaucomatous damage [67,68,69,70].

With respect to the surrounding sclera, the lamina cribrosa has a limited structural integrity and is therefore particularly sensitive to shifts in the TLCP gradient [71]. As TLCP increases, the connective tissues of the lamina cribrosa are compressed and retreat outwards towards the retrobulbar space, adopting an “excavated” appearance with time (Figure 3). However, chronic elevation of TLCP will eventually degrade the collagenous fibrils of the lamina cribrosa, inflaming its excavated appearance and reducing the distance between the intraocular and retrobulbar reservoirs [72]. Given the relationship between distance and the magnitude of a pressure differential, this thinning of the lamina cribrosa will permanently elevate the TLCP gradient within a glaucomatous eye [73]. Consequently, more severe glaucoma cases generally continue to progress even after surgical or pharmacological intervention that returns IOP back to normal levels. Similarly, myopia-related stretching of the scleral membranes is also known to enhance glaucoma risk. This further reflects the critical role that lamina cribrosa thickness plays in TLCP maintenance [74,75].

In the short term, enhanced TLCP gradients contribute to the degeneration of RCGs by inhibiting orthograde and retrograde axonal transport. This severely impairs the movement of vesicles, neurotransmitters, neurofilaments and organelles—most importantly the mitochondria—from the retina to the lateral geniculate nucleus [1,76]. Additionally, long-term exposure to high TLCP will starve the RGC axons of nutrients, leading to axonal degeneration and eventually apoptosis [77]. This pathogenic cascade is known to be initiated by elevated IOP, which is why it has been difficult to piece together the underlying mechanism by which normal-tension glaucoma (NTG) develops. It is thought that patients with NTG may have mutations in the gene encoding optineurin, a protein that regulates membrane trafficking, endosomal transport and autophagy through interactions with Rab8, myosin V1 and Huntingtin [78].

During the progression of glaucoma, the arterial blood supply that serves the retina tends to be unaffected by changes in pressure across the lamina cribrosa. Only during onset of acute angle closure glaucoma, in which IOP reaches as high as 60–70 mmHg, can blood flow into the retinal arteries become restricted. On the other hand, venous return of blood from the eye is very sensitive to shifts in IOP and TLCP, as the hydrostatic pressure of veins is much lower than that of arteries. According to models informed by the Starling resistance mechanism, the pressure gradient across the endothelium of retinal veins is stretched across the entire length of the vessel. Consequently, venous pressure in the prelaminar region tends to be lower than the IOP, making the central vein of the retina particularly susceptible to glaucoma-induced occlusion [77].

### 3.2. Mechanisms of IOP Elevation

Chronic IOP elevation is often established through a blockade of the pathways involved in AH outflow. Most commonly, an enlargement of the lens or ciliary body will mechanically block the anterior angle and prevent the re-absorption of AH fluids by the TM [79]. This “ciliary block” mechanism can also be induced by shifts in lens positioning and thickening of the hyaloid canal [77]. Furthermore, the TM can also be shielded by a distention of the peripheral iris tissue. During this “pupil block” mechanism, a substantial pressure gradient is built up across the surface of the iris, promoting the peripheral portions of this tissue to bow outwards and block the anterior angle. For an excellent description of the biophysical mechanisms by which ciliary block and pupil block occur, see Sun and Dai’s *Medical Treatment of Glaucoma*. Finally, closure of the anterior angle can also be mediated by small changes in choroidal volume, which do so by altering the morphology of the iris [80].

Given that the rate of AH production is not significantly altered in glaucoma [26], and that the uveoscleral pathway drains only 10% of AH fluids, increased trabecular resistance is often elaborated as a primary mechanism by which IOP elevates in POAG. This change is thought to be mediated by fibrosis and cell death in the pre-glaucomatous TM, [81] processes that cannot yet be detected in clinical practice. Reduced ECM turnover that occurs with age also contributes to changes in the resistance of the meshwork structure. We will further consider the molecular details of these pathways in the next section.

Several genetic risk factors have also been linked with enhanced trabecular resistance, especially among patients with juvenile-onset, open angle glaucoma. Mutated myocilin, a protein involved in membrane trafficking within TM cells, can disrupt mitochondrial function [82] and accumulate in the meshwork ECM [77], leading to reduced AH outflow. Similarly, LOX1 mutations can lead to the detachment of basement membrane flakes from the ciliary body and iris [83]. These loose fibers often go on to block the TM and cause glaucoma.

Additionally, multiple internal and external factors can also affect trabecular resistance. In young myopia patients, pigments released from the posterior iris through eye rubbing can be deposited in the trabecular region. Here, pigment particles block conventional AH outflow pathways and lead to elevated IOP [84]. Moreover, in the aphakic eye (an eye without a lens), angiogenic and growth factors released from the retina can sometimes inadvertently reach the TM and mediate changes in its absorptive properties. A similar progression is often observed in patients following cataract surgery [85], which can be attributed to transient reductions in postoperative lens resistance. Inflammatory materials from uveitis can also trigger changes in TM resistance that can lead to the progression of inflammatory glaucoma [86].

Finally, certain ocular traumas can raise IOP by cleaving the TM from the scleral spur and disrupting intraocular homeostasis. Steroid eye drops can also change the resistance properties of the TM by changing its surface ECM profile [87]. Notably, steroid drops are used in some animal disease models to induce glaucoma pathogenesis.

## 4. Trabecular Meshwork and Tissue-Engineered Models

### 4.1. Substructures in the TM

The skeletal architecture of the TM is primarily supported by a core network of collagenous and elastic fibers, called lamellae (Figure 2). The lamellar core is separated from the TM parenchyma by a basal lamina sheet, across which some fibers extend to form a porous matrix [8]. Within this porous structure, the TM can be subdivided into three histological layers: the uveal trabecular meshwork (UTM), the corneoscleral trabecular meshwork (CTM) and the juxtacanalicular tissue (JCT). The UTM consists of 1–3 layers of lamellae and directly interfaces with the AH. By contrast, the CTM, which is situated directly beneath the CTM, is about 8–15 lamellae thick. These two layers work together as an epithelial exchange surface to phagocytose and filter the soluble pigments within the AH [88]. The JCT is positioned between the CTM and the endothelial wall of SC, and unlike the other two TM layers, it does not consist of lamellae. Rather, it is a hyaluronic and proteoglycan-enriched matrix of connective tissue housing several layers of mesenchymal cells and elastic fibers. The latter material aggregate within the JCT to form the cribriform plexus, an elastic structure which critically regulates AH outflow in response to IOP variations [89]. Given that the pores of the UTM and CTM are far too large to adequately resist AH outflow, it is believed that trabecular resistance is primarily mediated by the structures of the JCT [8].

### 4.2. Changes of the TM in Glaucoma

#### 4.2.1. Excessive Deposition of ECM in the Pre-Glaucomatous Eye

Under normal conditions, the ECM remodeling process in the TM is thought to proceed continuously, mediated by the high expression levels of matrix metalloproteinase 2 (MMP2), tenascin C and α-smooth muscle actin (αSMA) in TM cells [90]. As mentioned earlier, the relatively high rate of ECM turnover in the TM compared to other adult tissues allows outflow facility to remain constant, even as the intraocular space undergoes transient variations in pressure. Furthermore, a chronic elevation of IOP for several hours or days is known to influence the expression patterns of ECM components and MMP activity to facilitate a more robust outflow response [83].

However, preceding glaucoma pathogenesis, the elastic cribriform plexus will often undergo several changes in structure and function. Most obviously, layers of sheath-derived plaques containing fibbrillin-1 and microfibrillar associated protein-1 will be deposited around the JCT, further blocking outflow facility, and leading to a net elevation of IOP. This pattern of fibrotic deposition is described as an endothelial to mesenchyme-like transition, a trans-differentiation process mediated by several molecular mechanisms [91]. It is postulated that TM cells that have undergone the endothelial to mesenchyme-like transition deposit excessive extracellular matrix in the juxtacanalicular tissue (JCT), increasing the hydraulic resistance to trabecular outflow. In addition, it may be possible that mesenchyme-like cells act as pericyte-like cells to cover the endothelium to reduce its permeability, increasing fluid resistance. Although the endothelial to mesenchymal transition of Schlemm’s canal endothelial cells was also observed in vitro, its mechanism and contribution to glaucoma progression in vivo is not yet clear [92].

Transforming growth factor-β (TGF-β) is believed to be the most important factor involved in this endothelial to mesenchyme-like transition. Under normal circumstances, TGF-β2 activity promotes matrix production in the TM as a means of maintaining intraocular homeostasis. This soluble factor is released as an inactive, latent protein complex that travels through the AH and associates with microfibrils in the ECM of TM cells. Here, the inactive complex undergoes proteolytic cleavage by activator proteins—includes plasmin, MMP-2, MMP-9 and thrombospondin-1 (TSP-1)—to liberate the active peptide [93]. Activated TGF-β2 molecules can then bind receptors on the surface of TM cells and enhance the production of connective tissue growth factor (CTGF), which in turn promotes the deposition of fibrotic plaques within the JCT [94]. For unknown reasons, the AH of a POAG patient has significantly elevated levels of TGF-β2 [83]. Moreover, infusing exogenous TGF-β2 into organ cultures is known to induce glaucomatous changes by impairing the rate of trabecular outflow [95,96]. Taken together, these provocative findings suggest that TGF-β2 likely plays a major role in glaucoma-related fibrotic changes. However, further investigation is required to disentangle the properties of TGF-β2 signaling the in healthy and glaucomatous eye [97].

The characteristic fibrosis associated with glaucoma may also be mediated by a disruption of redox homeostasis within the TM cells [98]. Peroxiredoxin 6 (Prdx6), a moonlighting antioxidant which maintains levels of reactive species (ROS) within the cytosol, decreases in aging and glaucomatous TM cells. Such changes disinhibit the production of ROS, which in turn enhances the expression of TGF-β and several ECM genes, including α-SM and fibronectin. High concentrations of ROS also lead to DNA damage, and the subsequent loss of TM cells. In this way, the progressive reduction of Prfx6 expression in the TM gradually enhances the resistance of this tissue to AH outflow, contributing to the elevation of IOP [99].

Recent studies have also indicated a link between ECM deposition and TM cell dysfunction. The endoplasmic reticulum (ER) is involved in the synthesis and folding of secreted ECM proteins. In the glaucomatous eye, the rapid rate of ECM deposition places stress on the ER’s synthetic machinery, thereby increasing the incidence of protein misfolding. This will activate the unfolded protein response (UPR) in an attempt to support the biosynthetic functions of the ER. However, chronic use of this response will eventually lead to further ER dysfunction and cell death [100,101].

#### 4.2.2. Changes in Cell Volume Regulation and Cytoskeletal Integrity

Regulation of the cytoplasmic volume is essential to maintaining a cell’s integrity during morphology changes, proliferation and migration. The volume of TM cells is kept relatively stable by an osmotic pressure gradient across both sides of its cell membrane. This gradient is in turn maintained by large conductance calcium-activated potassium channels (BKCa) and volume-regulated anion channels (VRAC), both of which vary the volume of TM cells to negotiate shifts in outflow facility [102]. Patients with open-angle glaucoma often display reduced VRAC activity, suggesting that impairments in the regulation of cell volume may also contribute to the IOP dysregulation that precedes POAG.

Besides cell volume, improper regulation of the mechanical and contractile properties of TM cells is also thought to enhance resistance to AH outflow. There is growing evidence that contraction of the TM actually reduces outflow facility, while TM relaxation promotes AH uptake. This contractile behavior is critically modulated by the Rho/ROCK signaling pathway [103]. When bound to GTP, the Rho GTPase will activate Rho-associated kinase (ROCK) along with several other downstream effectors proteins to trigger TM contraction. Rho/ROCK signaling is also likely to participate in the endothelial to mesenchyme transition of TM cells, as this pathway is also strongly activated by the Smad-independent TGF-β2 pathway [104].

Based upon these observations, it was long hypothesized that the Rho/ROCK pathway was overactive in glaucomatous TM tissue. However, to our knowledge, there is still no evidence supporting this perspective. Interestingly though, inhibiting the Rho/ROCK pathway has been identified as an efficient means of reducing outflow resistance and lowering IOP. In fact, multiple Rho inhibitors, including Y-27,632 and K-115, have shown early success at promoting RGC survival and regeneration in animal models [103,105].

### 4.3. Tissue Engineered Models for Trabecular Pathway Study

TM cells were first isolated to explore the cellular and molecular basis of AH outflow in 1979 [106]. Using rudimentary tissue–culture assays, the physiology and gene expression of primary TM cells was thoroughly studied, leading the identification of the first glaucoma-related gene—myocilin [107]. Since then, investigations of the trabecular outflow pathway have made use of several in vitro or ex-vivo models, including 2D cell cultures of TM or SC endothelial cells, whole eyes and anterior segmentation models. However, despite the initial utility of these models in advancing the field of glaucoma biology over the past century, each has important limitations that must be taken into account.

2D-cell-culture models are primarily limited in their ability to sustainably recapitulate the physiology of the TM and SC. While the lifespan of primary TM cells depends directly upon the donor age and culture conditions, these cells are generally only usable until passage five, even under the best circumstances. Consequently, transformed TM cell lines were generated in the 1990s to be more sustainable and phenotypically stable. However, the physiology and gene expression profiles of these cell lines differed greatly from that of primary cells, and they were not favored for use in 2D-culture models.

Currently, due to the potential for sample contamination during primary cell recovery and gene expression changes after multiple passage cycles, the identity of primary TM cells generally needs to be verified. Characteristic markers of physiologically relevant TM cells include myocilin (MYOC), matrix gla protein (MGP), caveolin 1(Cav 1), collagen 4 alpha 5 (Col4A5), tissue inhibitor of metalloproteinase 3 (TIMP3), αβ-Crystallin and smooth muscle actin (SMA). Additionally, the successful upregulation of MYOC through a steroid treatment or phagocytosis assay are also used to confirm the trabecular phenotype [108]. Similarly, primary SC endothelial cells are relatively difficult to maintain in culture, and there is currently no commercially available supply of this cell type. Identify verification is also recommended to ensure the physiological relevance of cultured SC cells, and can be accomplished by confirming the presence of fibulin-2, VE-cadherin and integrin-α6 as well as the absence of LYVE-1 [108]. Most importantly, although 2D cell culture is the most common and inexpensive approach to trabecular study, it cannot fully mimic the 3D structure, cell–ECM interactions and fluid environment of the intraocular space.

Two major advantages of whole eye models are that the globe structure is fully intact, and all cell types are situated their normal ECM microenvironment. However, due to the absence of both vascular and AH perfusion, trabecular cells in whole eye models die within 36 h. Additionally, some animal models exhibit a time-dependent “washout effect”, in which the cells and ECM along the conventional pathway detach from their original sites. This gradually diminishes TM resistance to AH outflow [109].

In contrast with whole eye models, organ culture anterior segments (OCAS) recovered from humans or another animal species postmortem are able to sustain AH and hematologic perfusion indefinitely. Instead of separating the anterior chamber and posterior chambers, the eyes being preserved as OCAS models are cut at the equator and the iris, lens, choroid, ciliary body and vitreous humor are removed. To eliminate the confounding effects of interindividual variations in ocular structure, OCAS models of two eyes from the same donor are typically perfused together, allowing one of them to serve as an experimental control. In this way, OCAS models have provided the most physiologically relevant means of examining aqueous outflow for nearly 20 years. However, human OCAS pairs tend to be very expensive to obtain and limited in supply, considering the large quantity needed to complete a single experiment. Furthermore, although animal source of OCAS are plentiful, inevitable species difference in ocular physiology limit their utility [108].

In the face of these challenges, tissue engineers have begun developing novel models of the optic globe that better recapitulate trabecular physiology and glaucoma pathogenesis.

#### 4.3.1. 2D Topographical Scaffold Models of the TM

One of the primarily phenotypic changes induced by culturing TM cells on 2D plastic flasks is a downregulation of myocilin [110]. This can likely be attributed to a mismatch in the topographical architecture of the lamella-based TM in vivo and the smooth surface in vitro culture platforms. Consequently, multiple studies have used microfabrication or electrospun nanofiber-based methods to build topographically accurate scaffolds for TM models [111,112,113] (Figure 4A). Russel et al. (2008), used soft lithography to fabricate a novel nanopatterned polyurethane surface finished with a ridge-and-groove pattern. Interestingly, the primary TM cells cultured on this novel platform took root along the entire surface of the anisotropic pattern, preventing them from aggregating into clumps as they do on planar flasks. Furthermore, the use of this platform caused cultured TM cells to adopt a healthy, elongate morphology and restored myocilin expression back to physiological levels in most cells that were assessed [112].

Building off these findings, Kim et al. (2011), compared the behavior of TM cells cultured on four distinct poly(etherurethane)urea (PEUU) surfaces. Each of these platforms of varied in how they were manufactured (Electrospun nanofibers or soft lithography) and in the topography of their ridge-and-groove surface pattern (either oriented or random). Myocilin expression of cells cultured on both topographical surfaces increased substantially with respect to those cultured on the planar control surface. However, the cells cultured on the randomly patterned, electrospun nanofiber-based surface expressed more myocilin than those cultured on the oriented, electrospun nanofiber-based surface, which in turn expressed higher myocilin levels than the oriented, soft lithography-based surface [111]. The geometric attributes of other micro- and nanotopographically contoured PDMS surfaces were also shown to enhance the porosity of cultured TM networks [113], cell adhesion, proliferation and migration [117] (Figure 4A). Taken together, the literature suggests that some of the in vivo characteristics of TM cells can be retained when cultured on topographically contoured surfaces. Unfortunately, many of the in vivo properties of TM cells that are not recapitulated in 2D-culture cannot be influenced by topographical factors. This includes the expression of αB-crystallin, another characteristic trabecular protein and the sensitivity of TM cells to dexamethasone, a potent glucocorticoid [112].

Beyond ridge-and-groove patterning, the surface topography of 2D-culture platforms can also be varied through the use porous membranes. As early as 1988, TM cell monolayers were cultured on porous filters as a means of quantifying the tissue’s hydraulic conductivity [118]. More recently, Professor Yubing Xie’s group used lithography techniques to develop a highly porous, gelatin-coated membrane of SU-8, a biocompatible and photo-definable epoxy [114,119,120,121] ((i) of Figure 4B). Similar to previous platforms, culture of TM cells on this membrane recapitulated the elongate in vivo morphology of this cell type. Compared with polyester Transwell inserts—a commercially available membrane with lower porosity and a less regular porous structure—the SU-8 membrane promotes higher cell coverage and more elongate cell morphologies. Moreover, use of the SU-8 culture surface upregulated not only myocilin, but also αB-crystallin, α-SMA, collagen IV and fibronectin in cultured TM cells. Taken together, these findings emphasize the significance of porosity and membrane pore pattern in the assembly of physiologically relevant trabecular models.

Given its utility, several groups have mobilized the SU-8 membrane to generate both steroid-induced and TGF-β2-induced models of trabecular resistance in glaucoma. Torrejon et al. (2016 & 2018) went a step further and used the former model to further investigate the effects of ROCK inhibition on outflow facility. As mentioned earlier, steroid-induced changes in the endogenous TM are similar to those observed in primary open angle glaucoma (POAG). Specifically, both are characterized by enhanced fibrosis of the JCT, degeneration of the intra-trabecular space, upregulation of myocilin, cytoskeletal rearrangement, inhibition of phagocytosis and increased outflow resistance. The application of prednisolone acetate, a common ophthalmic corticosteroid, to an TM model cultured on an SU-8 membrane was shown to induce *each* of the glaucomatous changes described above. These data demonstrate that the steroid-induced model of trabecular resistance faithfully recapitulates the pathophysiology of glaucoma [119].

Furthermore, administration of the ROCK inhibitor Y27632 to the steroid-induced model was shown to attenuate the expression of myocilin, collagen IV and fibronectin, while increasing that of αB-crystallin [119]. Time-dependent shifts in the expression of MMPs, IL-α (known to induce MMP production), tissue inhibitor of metalloproteinase-1 (TIMP-1) and TGF-β (known to alter ECM metabolism, suppress MMP-3 and enhance fibronectin deposition) were also observed [114].

The ability of TGF-β2 to induce glaucoma in an SU-8-based TM model has also been studied. Following TGF-β2 treatment, the expression of myocilin, collagen IV, fibronectin, laminin, tissue transglutaminase (an enzyme that crosslinks ECM proteins to protect them from proteinases) and plasminogen activator inhibitor (PAI, an MMP inhibitor) in the TM cell layer increased. Moreover, the fibrotic and cytoskeletal changes to the TM known to be associated with TGF-β2 overproduction in vivo are recapitulated when the model is perfused with exogenous TGF-β2. Treatment of the TGF-β2-induced glaucoma model with Y27632 was shown to promote cytoskeleton reorganization and reduce the expression of both ECM and myocilin, indicating a potential interaction between the ROCK and TGF-β2 pathways [119]. Taken together, these findings confirm the therapeutic potential of ROCK inhibition and illustrate the inherent plasticity of SU-8-based, steroid-induced models of glaucoma.

Membranes with porous topologies have also been used to model the behavior of SC cells in vitro. In 2011, Pedrigi et al. used porous membranes to construct a confocal time-lapse face imaging system with which to study the giant vacuoles of SC ((i) of Figure 4C). These structures are outpouchings of the SC endothelium that bulge into the canal lumen, leaving fluid-filled cavities between the cellular and basement membrane [115]. To manufacture their model, Pedrigi et al. seeded SC cells onto porous Transwells bathed in media reservoirs, which they maintained at several distinct pressure levels throughout the experiment. The giant vacuoles–like (GVL) which eventually formed very closely mimicked the “signet ring” appearance of the in vivo vacuolar architecture [115] ((ii) of Figure 4C). Furthermore, the functional attributes of the GVL structures in this model were shown to be pressure-dependent. Although large variation existed between individual samples, increases in fluid pressure were typically associated with larger GVL surface areas and thinner cavity walls. Additionally, outflow resistance across the GVL structures was also pressure-dependent, although this parameter did not exhibit much interindividual variation. Notably, the GVL structures that formed in this Transwell model were significantly larger than those found in vivo. One potential explanation for this is that the authors may have overestimated the actual pressure gradient to which endogenous SC cells are exposed. This finding may also simply reflect the substantially higher surface area of SC cells cultured in vitro, compared to those found in vivo.

SU-8 scaffolding has also been used to model the fluid dynamics across the walls of SC. Instead of using a gelatinous membrane, the SU-8 scaffold was coated in Extracel (HA-Gelin S) to better maintain the proliferative behavior and cytoskeletal architecture of SC cells. This Extracel coating also allowed SC cells to maintain physiological expression levels of fibulin-2, CD31 and VE-cadherin, all of which tend to downregulate in traditional 2D-culture ((ii) of Figure 4B). The application of a pressure gradient across the cultured monolayer also induced the formation of paracellular and transcellular vacuoles, hallmarks of in vivo SC cell physiology. In 2015, Dautriche et al. used this model to demonstrate that overexposure to TGF-β2 also induces SC cells to undergo an endothelial to mesenchyme transition very similar to that observed in the TM cells of the glaucomatous JCT. This finding further conveys the plasticity and physiological relevance of Extracel-based SU-8 scaffolds in the modeling of SC [92].

SC cells cultured on this surface also exhibited an unusually high transfection efficiency when challenged with exogenous siRNA, suggesting that in vivo SC cells may be similarly susceptible to therapeutic transfection [92]. Intriguingly, Tian et al. (2020) recently observed that, when exposed to shear stress and exogenous VEGF-C, SC-like cell monolayers can be differentiated from human adipose-derived stem cells (ADSCs) co-cultured with TM cells on a micropatterned SU-8 scaffold [122]. This finding has the potential to dramatically expand the availability of accurate trabecular outflow models, although the phenotypic characteristics of SC-like cells must be further studied to ensure their analogy to their in vivo counterparts.

##### 4.3.2. Three-Dimensional Scaffold for TM Models

Although several studies demonstrate the utility of SU-8-based scaffolds in recapitulating the morphology, gene expression and cytokine responses of TM and SC cells, the 2D nature of these platforms restricts the overall trabecular thickness to about 20 μm. This does not allow researchers to observe cell migration or cell–ECM interactions in three-dimensions, both of which play an important role in glaucoma pathogenesis. Hence, multiple groups have attempted to establish more physiologically relevant, 3D hydrogels of cultured TM cells [116,123,124].

Hydrogels are networks of crosslinked, hydrophilic polymers often used to recapitulate the 3D architecture of organ systems in tissue engineered models. These materials are so useful in cell culture because they provide a biocompatible, degradable, hydrated microenvironment that mimics the cell–ECM interactions of natural tissues. A notable limitation of hydrogel-based cell culture is that it is often very difficult to reestablish a microvascular infrastructure that can support nutrient and oxygen transport inside the 3D hydrogel. Fortunately, this is not a problem for TM—an avascular tissue which instead relies on the circulation of soluble factors in the AH fluids. This endogenous exchange process can be easily mimicked in vitro [125].

The two primary characteristics of 3D TM models which we will consider here are the porous structure and mechanical strength of the hydrogel material used. The pore size and shape of a hydrogel will dramatically influence cell attachment and migration, as well as nutrient and oxygen transport within the hydrogel [37,117]. Additionally, TM cells cultured on polyacrylamide hydrogel surfaces with varying tensile strengths exhibit distinct morphologic features and drug responses, suggesting that the mechanical properties of a hydrogel critically inform cellular behavior [126].

In a study by Osmond et al. (2017), a collagen scaffold with an aligned porous structure was synthesized by unidirectional freezing and lyophilization [123]. To better mimic the in vivo architecture, they also treated the collagen scaffold with chondroitin sulfate (CS) to chemically link glycosaminoglycan residues to its surface. These mucopolysaccharides functionally contribute to the filtering action and trabecular outflow resistance generated at the TM interface. Measured via atomic force microscopy (AFM), the mechanical strength of both the collagen and collagen-CS-treated scaffolds were very similar to that of the trabecular connective tissues. Furthermore, the expression of myocilin increased in TM cells cultured on both the collagen and collagen-CS scaffolds 2 weeks after the initial cell seeding, although that of cells cultured on the collagen-CS surface was slightly higher. TM cell proliferation and migration was also observed on both scaffolds, demonstrating the utility of this platform in modeling the plastic properties of the trabecular outflow pathway [123].

Recently, Waduthanthri et al. (2019) developed an injectable peptide hydrogel for use in TM tissue engineering, called MAX8. Each MAX8 peptide is a 20 amino acid-long chain made up of alternating hydrophobic and hydrophilic residues ((i) of Figure 4D). To boost the material’s biocompatibility, each peptide is flanked on either side by a tetra-peptide GRGD (Gly–Arg–Gly–Asp) sequence, which mimics the RGD (Arg–Gly–Asp) motif of cellular integrins and enables interactions between the MAX8 peptide and several ECM components. TM cells seeded on a solidified MAX8 hydrogel were found to exhibit morphologic, gene expression and migratory properties analogous to those observed in vivo. In particular, the high migratory capacity of TM cells cultured with MAX8 allowed them to spread out across the entire scaffold within only 7 days ((ii) of Figure 4D). This observation likely reflects the incorporation of GRGD flanking sequences in the MAX8 peptides, whose biocompatibility dramatically enhances the incidence of cell-scaffold interactions. The authors also build a perfusion model of trabecular outflow using a MAX8 hydrogel. When treated with dexamethasone, this model exhibited a time-dependent enhancement of trabecular hydraulic resistance, which was measured by quantifying the pressure differences across the modeled TM. Briefly, the authors built a perfusion system to circulate media through their 3D model of trabecular physiology at a constant flow rate. They measured the pressure differences across the TM to calculate the hydraulic resistance or “trabecular resistance,” of the model in different conditions [116]. These findings again reflect the physiological relevance and plasticity of MAX8′s rheological properties and the use of this material in therapeutic implants and bioprinted 3D models of TM physiology should be further considered [116].

In a similar vein, Bouchemi et al. (2017) built a TM model using Matrigel^®^, a basement membrane matrix secreted by Engelbreth–Holm–Swarm (EHS) mouse sarcoma cells. Matrigel^®^ is widely used in tissue engineered cultures to model the cell–ECM interactions involved in cancer metastasis, angiogenesis, cell migration and differentiation [127]. In stark contrast to the uniform spreading of TM cells observed in MAX8-based hydrogels, primary TM cells cultured in Matrigel^®^ were found to aggregate into large clusters after 11 days. Similar to other 2D and 3D models, treatment of the Matrigel^®^ model with dexamethasone and TGF-β2 induced glaucomatous changes in TM cell morphology, ECM expression and F-actin nucleation [124]. After confirming its physiological relevance, the authors used their Matrigel^®^-based model to assess the toxigenic effects of benzalkonium chloride (BAK), a preservative found in many anti-glaucoma drugs. They observed a time-dependent increase in IL-6 and IL-8 expression in cultured TM tissue treated with BAK, indicating that this preservative may actually induce glaucomatous changes in AH outflow. This response is likely mediated by BAK-induced oxidative stress, as the application of exogenous H_2_O_2_ on 3D-cultured TM cells induces the same inflammatory phenotype. Notably, while similar increases in IL-6 and IL-8 production were observed in 2D-culture and 3D-Matrigel^®^ models treated with BAK, the TM cells in the latter model exhibited an even greater production of inflammatory cytokines. This perhaps speaks to the true magnitude of BAK-induced trabecular inflammation in vivo. BAK signaling was also shown to downregulate MMP-9, which in turn reduced ECM remodeling and contributed to the elevation of IOP [124].

Apart from using hydrogel scaffolds, 3D tissue engineered models of trabecular physiology can also be established using a 3D printer. Bioprinting represents an emerging technology that is growing in popularity due to the rapid manufacture and well-controlled geometry of bioprinted materials. Compared with standard lithographic methods, 3D bioprinting can produce more nuanced architectural patterns on a wider array of biomaterials. However, despite its clear advantages over other methods, the resolution of extrusion-based 3D printing technology is not yet sufficient to reproduce the 10-micron-thick pores of the in vivo TM. Consequently, there has been very limited progress towards establishing 3D bioprinted models of trabecular outflow. Recently, Huff et al. (2017) worked to optimize the pore resolution of these models by using sodium alginate and methacrylated gelatin (GelMA) bioinks, although more work must done to determine the optimal printing parameters [128]. Compared with extrusion-based 3D printing, Stereolithographic 3D bioprinting may be a more promising method for building future TM in vitro model because its higher resolution and the absence of mechanical extrusion.

## 5. Tissue-Engineered Models of Retinal Ganglion Cells

### 5.1. Molecular Mechanisms of Retinal Ganglion Cell Death in Glaucoma

The death of RGCs is a crucial stage in the pathogenic progression of glaucoma, and is generally induced by a failure of axonal transport, deprivation of neurotrophic factors, the activation of intrinsic and extrinsic apoptosis signals, mitochondrial dysfunction, excitotoxic damage, oxidative stress, reactive gliosis and a loss of synaptic connectivity [129]. We highlight two of these mechanisms below. It is important to note that while IOP is a significant risk factor for RGC degeneration, only a limited subset of individual with IOP outside of the normal range will develop glaucoma. Moreover, even after surgical or pharmacological intervention to lower IOP, a significant number of glaucoma patients still continue to experience loss vision. This strongly suggests that there are molecular or biophysical mechanisms other than IOP elevation that contribute to RGC degeneration. Therefore, disease models that further elucidate the mechanisms underlying RGC death in glaucoma are of great significance.

#### 5.1.1. Neurotrophins

Neurotrophins are diffusible tropic molecules that mediate key cellular responses during the development and maturation of the central nervous system (CNS). Furthermore, neurotropic factors have been shown to have potent anti-apoptotic effects on the nervous tissues of patients with neurodegenerative disease [130]. The most common neurotrophins that perform these functions include nerve growth factor (NGF), brain-derived neurotrophin factor (BDNF), neurotrophin-3 (NY-3) and neutrophin-4/5 (NT-4/5). These soluble peptides bind to tropomyosin related kinase (Trk) receptors and p75 receptors on the surface of adult neuronal cells, where they initiate signaling cascades involved in cell survival [131].

BDNF is strongly expressed in the superior colliculus, the synaptic target of many RGC neurons traveling into the brain. Generally, this factor is believed to contribute to the selective survival of RGCs during retina development, and is known to be upregulated in the early stages of axonal repair in the optic nerve [132]. Although the retina also expresses BDNF, it has been proposed that retina-derived BDNF is a supplementary reserve that can only temporarily support degenerating RGCs which have lost their connection to the superior colliculus [129]. Hence, it is believed that the death of RGCs in glaucoma is caused in part by IOP-induced axonal damage, which impairs the transport of BDNF from the superior colliculus to the RGC soma. This disease-causing mechanism has been the target of several novel therapeutic approaches, including injection of exogenous BDNF, viral mediated BDNF gene transfer and TrkB (receptor for BDNF) gene transfer. These methods have been shown to significantly enhance the survival of damaged RGCs [133]. Unfortunately, the duration of this effect is limited, and BDNF cannot stimulate axonal regeneration. NGF, ciliary neurotrophic factor (CNTF), and glial cell line-derived neurotrophic factor (GDNF) are also known to exhibit similar neuroprotective effects in the glaucomatous eye. CNTF may also be able to stimulate axonal repair following injury through JAK/STAT3-dependent signaling [134].

#### 5.1.2. Apoptosis Activation

Both intrinsic and extrinsic apoptotic mechanisms contribute to RGC apoptosis during glaucoma. The intrinsic pathway involves a complex interplay between anti-apoptotic and pro-apoptotic molecules within the mitochondria, some of which we highlight below.

The mitogen-activated protein kinase (MAPK) family is comprised of several proteins that critically regulate the intrinsic apoptotic cascade, including Erk1/2, c-Jun N-terminal kinases (JNKs), p38 Erk5 [135]. Erk1/2 is an anti-apoptotic molecule involved in BDNF-dependent survival signaling [136]. Conversely, JNK and p38 are responsive to stress signals and direct the expression of pro-apoptotic genes, as is thought to be the case following glaucoma-related axotomy or nerve injury. Upstream of both JNK and p38 is the apoptosis signal regulating kinase 1 (ASK1), a member of mitogen-activated protein kinase kinase kinase (MAPKKK or MAP3 K) family. ASK1 is thought to directly sense stressful stimuli, inflammatory cytokines and oxidative stress to initiate the intrinsic apoptotic pathway. Dysfunction of this protein may play a role in the progression of neurodegenerative disease [137]. Three genes in the Bcl-2 gene family are also through to play important roles in regulating RGC survival during glaucoma pathogenesis. Specifically, Bcl-XL is a potent neuroprotective gene, while Box and BH-3 are pro-apoptotic [138]. Additionally, in response to DNA damage, neurotrophic factor deprivation, oxidative stress, ischemia and excitotoxicity, the p53 tumor suppressor protein potently drives the intrinsic apoptotic pathway in RGCs. All of these signals tend to be integrated by the mitochondrion, which weighs the relative contributions of each and determines a cell’s fate [139].

When a mitochondrion decides to initiate apoptosis, cytochrome c in the electron transport chain will bind Apaf-1 and form the apoptosome. This structure will further recruit and activate caspase-9, which will in turn trigger a proteolytic cascade that activates caspase-3. Besides cytochrome c, other proteins released from the mitochondria during apoptosis include the second mitochondria-derived activator of caspases (SMAC), apoptosis-inducing factor (AIF), endonuclease G (EndoG) and the high-temperature-requirement protein A2 (OMI/HTRA2) [140].

Glaucoma is known to be associated with dysregulation of the metabolic and apoptotic functions of the mitochondria. This is thought to be related to glaucoma-induced ischemia of the retinal tissue, which gradually depletes RGC mitochondria of oxygen and other nutrients. Furthermore, the blockade of axonal transport caused by changes in trans-lamina cribrosa pressure may prevent mitochondria from appropriately localizing within glaucomatous RGCs. Damage to mitochondrial DNA induced by age-related shifts in redox homeostasis may also contribute. The metabolic dysfunction that results from these three changes in mitochondrial physiology reduces the amount of ATP available in the glaucoma-predisposed optic nerve [141]. Given that ATP binds to cell-surface purinergic receptors (P2X or P2Y) to regulate neuronal physiology in the normal eye, abnormal Extracellular concentrations of ATP caused by mitochondria dysregulation can be toxigenic to RGCs. For an excellent review of the metabolic vulnerabilities associated with glaucoma, see Inman & Harun-Or-Rashid et al. (2017) [142].

In the extrinsic pathway, diffusible apoptotic signals from the tissue microenvironment disrupt the balance of several pro- and anti-apoptotic factors in the cytosol to induce cell death. The most relevant Extracellular apoptotic signals to glaucoma biology are tumor–necrosis factor-α (TNFα), Fas ligand (Fasl) and TNF-related apoptosis-inducing ligands (TRAIL). Upon receipt of one or more of these signals, activated “death receptors” will recruit Fas-associated death domains and precaspase-8 molecules to form the death-inducing signaling complex. This complex will then mediate the autoproteolytic cleavage of precaspase-8 and the activation of the caspase-3 cascade [129].

TNFα is a pro-inflammatory cytokine activated from its membrane-bound precursor by an ADAM17-dependent cleavage mechanism. Müller glia are a potent source of TNFα in the eye, and this pro-apoptotic signal is known to be upregulated in the glaucomatous retina [143]. Moreover, the inhibition of TNFα signaling exerts neuroprotective effects on glaucomatous RGCs in vitro [144], although this treatment approach is likely not clinically useful due to the potential for immunosuppression. Similar to TNFα, Fasl is synthesized as a transmembrane protein and is cleaved from the surface of glial cells by MMPs. Interestingly, while soluble Fasl (sFasl) is neuroprotective, membrane Fasl (mFasl) is neurotoxic, and the two compete with each other for binding spots on RGCs [145]. Hence, the ratio between sFasl and mFasl, which in turn is determined by MMP activity, plays a key role in glaucoma pathogenesis.

### 5.2. Tissue-Engineered Models for the Study of Glaucomatous RGCs

Over the past several decades, insight into the glaucomatous degeneration of RGCs has primarily been gleaned from 2D cell cultures and animal models of retinal physiology. As with the study of TM cells, 2D-cultures of RGCs have been established from both primary and transformed RGC lines. While the former cells more closely mimic the retinal architecture, they do not survive well after several rounds of cell passage. Conversely, the latter cell lines are immortalized, exhibit rapid proliferation in vitro, and are easy to maintain in culture for long periods of time. However, the transformation of any cell line, including RGCs, generally produces cells whose expression patterns are substantially different from those of the endogenous tissue. Additionally, the culture of both cell lines as 2D monolayers cannot adequately recapitulate the natural cell–ECM interactions, biophysical properties or topography of the human retina. The use of animals, particularly murine models, can facilitate more physiologically relevant research of RGC degeneration [146]. However, ethical and financial concerns, time constraints, as well as interspecies differences in glaucoma pathogenesis limit the utility of these models.

To overcome the limitations associated with conventional RGC models, several tissue-engineered models of retinal physiology have been developed in recent years, including iPSC-derived organoids, topographically contoured 2D models and 3D hydrogels. A major advantage of these technologies is that they can be used not only to establish disease models and screen novel pharmaceuticals, but also to investigate the regenerative capacity of the NFL. As mentioned earlier, the utility of regenerative therapeutics in reversing glaucoma-related damage has been demonstrated in preclinical animal models, but not yet in humans. Progress still needs to be made in optimizing the transplant conditions of RGE allografts and xenografts into the human retina. Several tissue-engineered models, which we summarize here, have been developed with this aim in mind.

#### 5.2.1. Engineered 2D Scaffolds for RGC Culture

In contrast to the thick, porous, three-dimensional structure of the TM, the retina is a very thin, multilayered tissue. Therefore, 2D tissue engineered models can somewhat better mimic the in vivo architecture of the NFL (compared to that of the TM.) Despite this, 2D models cannot adequately recapitulate the differences in cell body size and retinal thickness that exist between the fovea and macula in vivo. Specifically, the RGC’s within the fovea tend to have much smaller cell bodies than those in the peripheral macula. Furthermore, while the foveal retina consists of about ten cell layers, its thickness gradually diminishes to form a monolayer in the peripheral regions of the macula. Unfortunately, it has been very difficult to mimic these anatomic distinctions in 2D-culture. Furthermore, the axons of RGCs cultured on planar surfaces grow in essentially random directions. This growth pattern is not constructive when attempting to model disease states or develop regenerative therapeutics, which require the successful migration of transplanted RGCs to the ONH.

To foster more ordered patterns of axonal growth, Kador et al. (2013) cultured murine RGCs on an electrospinning, poly-D, L-lactic acid (PLA) scaffold contoured with a radial pattern (a–d of Figure 5A). The radial topography of this platform served to orient RGC growth in the same direction as the electrospinning fibers. As expected, such a design gave rise to a radially oriented RGC network. Neurons cultured on this surface also displayed enhanced viability, longer axon lengths, and similar electrophysiological properties as controls cultured on a planar surface (Figure 5A(e,f)). Due to the random distribution of electrospun fibers at the center of the radial scaffold, RGC axons in this region fasciculate into axon bundles from which the radial patterned projected. Intriguingly, authors were able to affix the intact radial pattern of RGCs onto an explanted rat retina using Matrigel^®^ sealant. After 3–5 days, the graft was shown to precisely conform to the radial geometry of the endogenous NFL, suggesting that this PLA-based scaffold may be useful in the development of future regenerative therapies for glaucoma [147].

In 2014, Kador et al. further optimized this successful model by immobilizing Netrin-1, a guidance factor for RGC growth, onto the PLA surface. Building in a concentration gradient of Netrin-1 using UV-initiated crosslinking enhanced the radial polarization of cultured RGCs from 31% to 52%, further improving the technology’s suitability for therapeutic application [150]. It is likely that the glaucoma cell therapies of the future will mobilize such guidance factors and tissue engineered models to successfully reconnect RGC axons to the brain [151]. However, at present, there are still several problems associated with radial scaffold platforms that hinder their clinical utility. Most notably, axon distribution in the human NFL is not perfectly radial. Rather, some axons are arced or bowed in shape, especially at the fovea. Kador and his colleagues partly solve this problem through the use of inkjet 3D printing in a later study, but the precise organization of axons across an individual person’s retina cannot yet be fully mimicked in vitro [148] (Figure 5B). Additionally, the direct placement of an RGC scaffold onto the human retina during surgery is not realistic, creating the need for a foldable and biodegradable scaffold that can be easily handled intraoperatively.

Besides rodent RGC cells, RGCs derived from human-induced pluripotent stem cells (hiPSC) have also been used to generate 2D models of retinal physiology. In a study by Li et al. (2017), RGCs were differentiated from hiPSC and seeded onto a laminin-coated poly(lactic-co-glycolic acid) (PLGA) scaffold using electrostatic spinning. Prior to cell seeding, the expression levels of several RGC markers (tubulin, nestin, HuD, Tuj1, Thy1.1 and Brn3b), axon makers (neurofilament-light, neurofilament-medium and neurofilament-heavy) and voltage-gated sodium channels was assessed to verify the identity of the differentiated cells. Compared with cells seeded on coverslips, the RGCs on the PLGA scaffold had larger dendritic fields, enhanced dendrite complexity and augmented axon-like neurite outgrowth. Furthermore, no significant differences were observed in the electrophysiological properties of RGCs seeded on coverslips and scaffolds. The RGC scaffold was also successfully transplanted onto the retina of a rabbit and two rhesus monkeys [152].

An interesting 2D microfluidic platform was also recently developed by Wu et al. (2019). This group built a PDMS-based microfluidic chamber in which the hydraulic resistance could be varied to mimic the intraocular microenvironment (a of Figure 5C). RGCs were cultured on a flat surface inside the PDMA chamber, and the static pressure exerted on the monolayer was modulated by the height of the aqueous surface and the shape of the PDMS chamber. As the pressure was gradually increased, RGCs cultured in the chamber exhibited diminished neurite extension, axon length, total neurite length and dendritic branching [149] (Figure 5C(b,c)). Additionally, compared with traditional in vivo approaches, this microfluidic platform enabled real time observation and study of the NFL at the level of single cells. However, still missing from this model is a representation of in vivo cell–ECM interactions, which play a key role in the response of RGCs to mechanical stimulation.

#### 5.2.2. Three-Dimensional Hydrogel Scaffolds

Compared with 2D-cultures of RGC, 3D models better mimic the intraocular environment by promoting salient cell–cell and cell–ECM interactions. Due to the complex nature of retinal anatomy, very few 3D RGC models have been developed to date. Most of the available models make use of hydrogel scaffolding materials. The most straightforward way of establishing a 3D model of RGC physiology is by directly seeding RGCs onto a 3D scaffold. Hertz et al. (2013) assessed the survival and morphology of RGC and amacrine cells co-cultured directly on a library of hydrogels with different chemical makeups. By changing the ratio and molecular weights of poly(ethylene glycol) and poly(l-lysine), as well as the ratio of amines to hydroxyl residues on each polymer, the mechanical strength and chemical properties of the hydrogels were varied (a of Figure 6A). Of the many hydrogels that were produced, those with a 3:1 or 4:1 ratio of amines to hydroxyls in which high molecular-weight PEG was used optimized RGC and amacrine cell growth, regardless of the molecular weight of PLL that was used (b of Figure 6A).

Surprisingly, coating each hydrogel with laminin did not improve cell survival, likely because of the limited expression of MMP in cultured neurons. It was interesting see that the elastic modulus of the optimal hydrogels ranged from 3800 to 5700 Pa, which is much higher than that of in vivo ECM scaffolds (940–1800 Pa). These findings indicate that the chemical properties of optimal 3D hydrogels likely critically mediate RGC and amacrine cell attachment, survival and physiology. Although all cells were seeded on the upper surface of each 3D scaffold, multiple axons were observed to project deeply within the hydrogel matrix. Without any chemical or topographical guidance, the RGC axons did not align nor fasciculate into axon bundles in the 3D model, as in 2D-culture. Instead, all axons that penetrated the hydrogel were randomly oriented [153].

In 2016, Laughter et al. developed a biocompatible, injectable hydrogel with which to encapsulate RGC grafts being transplanted into the retinas of human glaucoma patients. This composite hydrogel, which they called PSHU–PNIPAAm–RGD, is comprised of three components: Poly(serinol hexamethylene urea) (PSHU), a modifiable backbone that mimics the native retinal ECM; Gly–Arg–Gly–Asp-Ser acid (GRGDS), an integrin/cell binding motif found in many components of the ECM; and poly(N-isopropylacrylamide) (PNIPAAm), a thermosensitive, water-soluble homopolymer which enables injection of the hydrogel at room temperature and gelation at 37 °C. The mechanical strength of the hydrogel is modulated by the degree to which GRGDS peptides cross-link to their cellular targets.

Compared with those cultured on 2D PDL-Laminin coverslips, RGCs cultured on PSHU–PNIPAAm–RGD were more viable and achieved a “laminar growth” morphology, meaning that they extended processes focused in one plane and formed synaptic connections through more direct patterns of growth. RGCs found in vivo show this laminar morphology as well, while those cultured on 2D scaffolds exhibit a more stellate morphology. The improved survival and axon extension behavior of RGCs cultured in PSHU–PNIPAAm–RGD can be attributed to the GRGDS sequences in this hydrogel, which can binds cellular integrin and more closely recapitulates the cell–substrate interactions that occur in the human retina [154] (Figure 6B).

3D hydrogel scaffolds can also be used to help differentiate RGCs from stem cells. Roozafzoon et al. (2015) studied the differentiation of RGCs from dental pulp stem cells (DPSCs) on both 2D-culture scaffolds and 3D fibrin gels. DPSCs are isolated from the dental pulp tissues of Sprague–Dawley rats, which form from neural crest cells during embryological development [155]. After five days, DPSCs cultured in an RGC differentiation medium begin to exhibit a multipolar morphology and increase their expression of several neuronal cell markers, including MAP2 and GFAP. Subsequently, FGF2 and Shh activate Pax6, an important neural/retinal progenitor marker and ATOH7, a crucial regulator of the RGC phenotype. This will initiate the RGC differentiation cascade [156]. After thirteen days of treatment with differentiation media, the expression of Pax6, Atoh7 and BRN3B in DPSCs cultured on a 3D fibrin gel increased 2.307-fold, 1.624-fold and 3.14-fold, respectively, compared to those cultured on 2D scaffolds. These findings indicate that 3D fibrin scaffolds can be useful in facilitating RGC differentiation, although further assays are needed to confirm the phenotype of the ganglion-like cells that are produced [157].

3D printing technologies have also been used to build models of the NFL. Kandor et al. (2013) developed a thermal inkjet 3D printing platform with which to build upon their previously described electrospinning nanofiber scaffold [147]. Mobilizing these technologies in conjunction with one another allowed the authors to simultaneously control the vertical positioning of cells in the synthetic NFL layer along with the patterning of RGC axons. This gave rise to a 3D model of retinal physiology whose structure mimicked both the uneven distribution of RGC bodies as well as the radial pattern of RGC axons observed in vivo [148]. Despite the incredible utility of such models in studying glaucoma pathogenesis, there are several notable problems associated with current 3D printing techniques. For example, inkjet printing intrinsically induces acute mechanical stress on printed cells, which will inevitably lead to some cell damage. Although certain printing buffers can be optimized to minimize the impact of such stresses on RGC survival, these mixtures often severely impair the attachment and electrophysiology of printed cells. Including alginate in the bioink to increase the viscosity and calcium content of the crosslinking buffer can only partially resolve these impairments. The in vivo environment may also help transplanted RGC recover from printing-induced stresses, although this cannot be confirmed because of the difficulty associated with transplanting for a large scaffold. Lastly, due to their limited viscosity and mechanical strength, bioprinted materials are not able to support multilayered structures. Therefore, printing techniques are not yet able to recapitulate the 3D architecture of complex tissue microenvironments. Hence, many improvements must be made to 3D printing technologies before they can be used to prepare clinically relevant models of the retina.

## 6. Conclusions

In this review, we summarized and compared the 2D and 3D tissue engineered models of glaucoma currently being used in ophthalmic research and regenerative medicine. In particular, we emphasized contemporary models that mobilize soft lithography, electrospinning, microfluidics, hydrogel scaffolding and 3D bioprinting to mimic trabecular and retinal physiology. After surveying the literature, we found that both 2D and 3D primary cultures of TM and SC cells effectively mimic the morphology and gene expression patterns of the JCT region and the inner wall of SC, respectively. Moreover, each tissue engineered model of conventional outflow is inherently malleable and can be induced to exhibit glaucomatous changes when exposed to steroids or TGFβ.

Most, but not all models, can also be influenced by variations in the pressure gradient to which the cell culture is subjected. The sensitivity of each model to such perfusion experiments is informed by the model’s structure as well as the strength of the biomaterials used in its construction. Despite the utility of current tissue engineered models in recapitulating the individual structures of the JCT and the inner wall of SC, no in vitro model has successfully examined the *joint* contributions of both structures to glaucoma pathogenesis. In the natural eye, the form and function of these two histological layers are inherently entangled, and TM–SC interactions play an important role in regulating outflow resistance. Therefore, future research should focus on establishing 3D models of conventional outflow that functionally integrate both TM cells and SC cells, possibly through the use of 3D bioprinting. These models would provide a promising platform for both mechanism study and drug testing in the future.

Similarly, both 2D and 3D tissue engineered models of the NFL provide a physiologically relevant microenvironment in which healthy RGCs can adopt their natural morphology, electrophysiological properties and pressure dynamics. However, no characteristics of the glaucomatous retina, such as axotomy or RGC apoptosis, could be recapitulated using these models. Additionally, the natural heterogeneity of the RGCs isolated from the human retina has been difficult to reconstitute using purified primary RGC in culture. In the future, co-culture of RGCs with supporting glial cells on 2D and 3D scaffolds may better mimic the in vivo anatomy and pathophysiology of the human retina. 3D bioprinting may also help establish models in which the cell density and thickness of RGC layers can be flexibly varied across the scaffold surface.

Moreover, a major limitation of current in vitro tissue culture models of glaucoma is that they only consider one aspect of retinal physiology. As we discussed throughout this review, glaucoma pathogenesis represents the failure of several ocular systems to adequately maintain the intraocular space, including the trabecular meshwork, the lamina cribrosa, the retrobulbar space, the ciliary body, the lens–iris interface, the retinal vasculature and the NFL. Relying on the dysfunction of only one ocular structure when manufacturing tissue culture models of glaucoma does not take into account the multifactorial nature of the disease. We therefore argue that future research efforts in ophthalmic tissue engineering should combine existing models of aqueous humor hydrodynamics and retinal degeneration in a single organ-on-a-chip platform. Such models will more closely mimic the complex architecture of the optic globe and help clarify how the dynamic interplay between multiple ocular systems contributes to glaucoma progression.

Studies to date involving tissue-engineered models of glaucoma have been focused on model design and verification of the TM and RGCs. However, despite their incredible utility, these models have not yet been fully combined in a system and applied to advance our mechanistic, quantitative understanding of glaucoma pathogenesis and develop clinically useful therapeutic strategies. Therefore, while we encourage the development of even more robust ocular models in the future, we also suggest researchers to make use of those already available to provide a more comprehensive, combined model to investigate the molecular mechanisms that underlie glaucoma. It is also critical that physician–scientists begin translating components of the tissue-engineered models described in this review into the clinic. In particular, we believe that TM and retinal implants which replace damaged or lost ocular tissues represent the next generation of glaucoma therapeutics. Preclinical and clinical trials should be focused on optimizing the thermosensitive properties of the injectable hydrogel carrier used in this therapeutic approach. In summary, tissue-engineered models may provide a unique platform to quickly screen novel therapeutics and mechanisms that can normalize intraocular pressure by normalizing TM structure and function and preventing RGC damages; to implant the engineered tissues to modulate in vivo function of the eye machineries.

## Figures and Tables

**Figure 1 micromachines-11-00612-f001:**
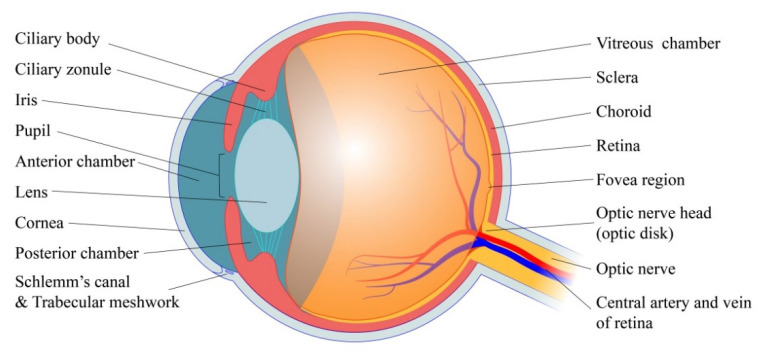
Physiology structure of the eye.

**Figure 2 micromachines-11-00612-f002:**
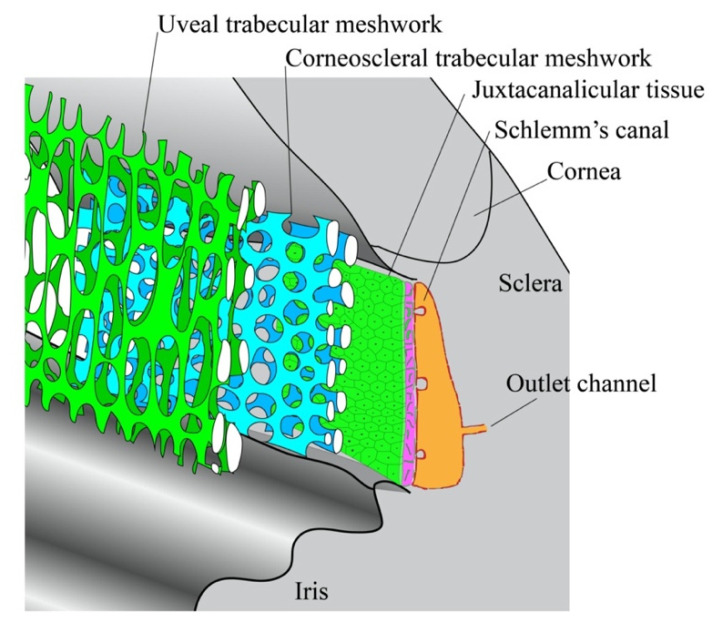
Substructures of trabecular meshwork and Schlemm’s canal.

**Figure 3 micromachines-11-00612-f003:**
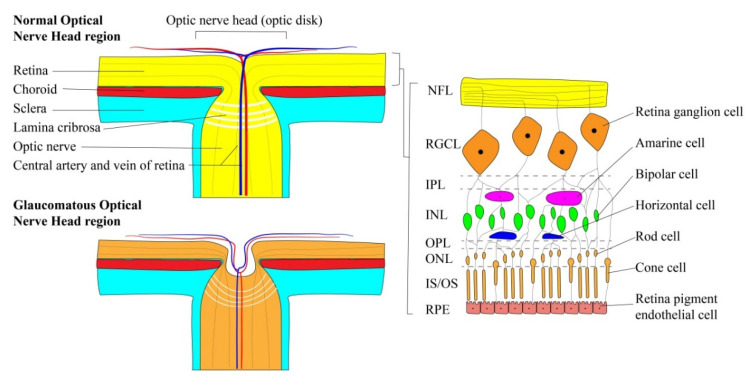
Physiological and pathologic structure of optic nerve head region. NFL: nerve fiber layer, RGCL: retina ganglion cell layer, IPL: inner plexiform layer, INL: inner nuclear layer, OPL: outer plexiform layer, ONL: outer nuclear layer, IS/OS: inner segment/outer segment, RPE: retina pigmented epithelium.

**Figure 4 micromachines-11-00612-f004:**
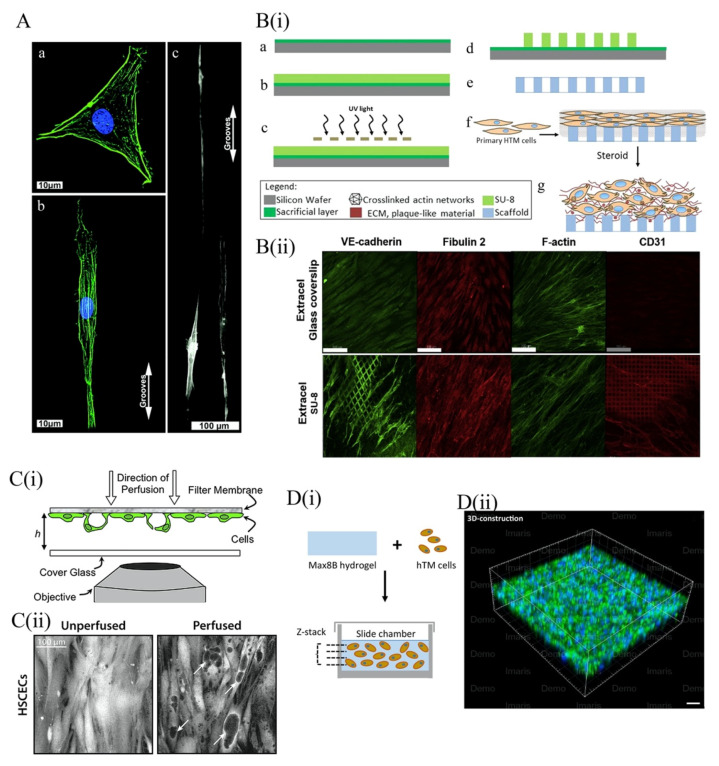
2D and 3D models of trabecular meshwork and Schlemm’s canal. (**A**) Comparison of TM cells on planar surface (**a**) and groove-patterned nano-surface (**b**,**c**). The actin filaments (green) were randomly oriented on planar surface but aligned on patterned surface. Blue: nuclei [112]; (**B**) TM and SC models based on porous SU-8 scaffolds; (**i**) fabrication of SU-8 scaffolds; (**a**) pre-cleaned silica wafer treated with a sacrificial layer; (**b**) photoresist SU-8 2010 coating; (**c**) UV-exposure using chrome mask; (**d**) post-exposure bake; (**e**) development to produce SU-8 freestanding scaffold; (**f**) HTM cell seeding on the SU-8 scaffold followed by 3D-culture; (**g**) steroid-treatment to generate glaucomatous 3D HTM model [114]; (**ii**) comparison of human Schlemm’s canal endothelial cells on glass coverslip and SU-8 porous scaffold. F-actin staining showed better fiber alignment on SU-8 scaffold. Expression cell characteristic marker CD31, which is lost in 2D-culture, was also recovered on SU-8 scaffold. Other characteristic markers—VE-cadherin and fibulin-2—were maintained. Scale bar is 100 μm [92]; (**C**) human Schlemm’s canal endothelial cells were cultured on Transwell; (**i**) diagram of the perfusion system; (**ii**) giant vacuole-like structures (arrows) observed with perfusion [115]; (**D**) (**i**) combining TM cells with Max8B hydrogel to reconstruct 3D environment; (**ii**) 3D-reconstruction of TM cells in MAX8B scaffold; green: F-actin; blue: nuclei red [116]. Figure republished with permission from each indicated reference ([112] for part A, [92,114] for part B, [115] for part C, [116] for part D).

**Figure 5 micromachines-11-00612-f005:**
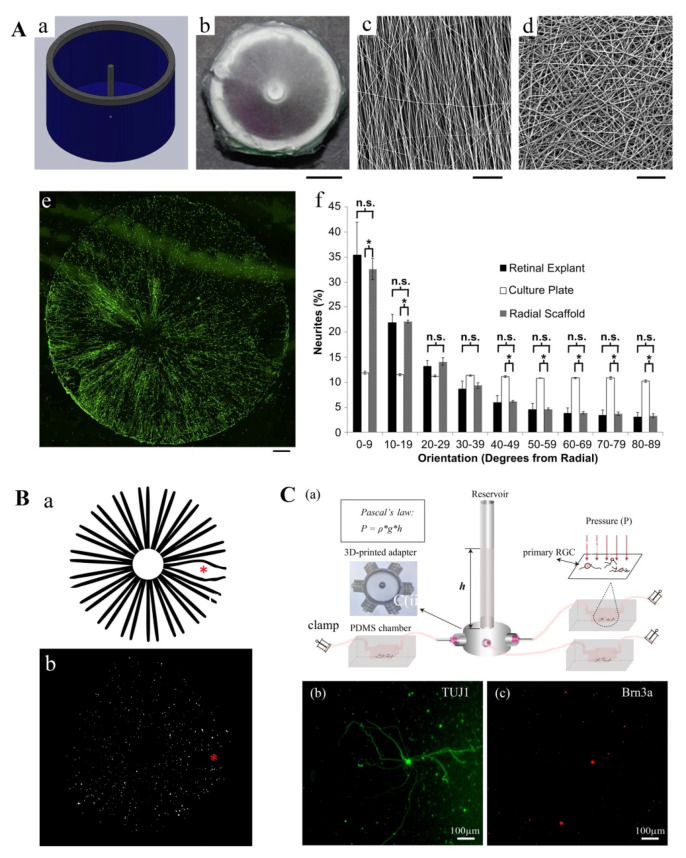
2D models of retina ganglion cell. (**A**) RGCs cultured on radial electrospun scaffolds mimic the axonal orientation of retina. Production and optimization of the radial electrospun scaffold. (**a**) Diagram of a 1.8-cm diameter radial collector containing a conducting central pole and rim grounded to the same source; (**b**) top view of an electrospun radial scaffold; (**c**,**d**) SEM image of peripheral radial fiber zone (**c**) and central random fiber zone (**d**); (**e**) fluorescence image of RGCs on radial scaffolds prohibited 81.1% ± 2.8 alignment of neurites in radial orientation. Green: β3-tubulin; (**f**) orientation analysis of RGC neurites on different scaffolds. No significant difference was observed between retinal explants and radial scaffold. Scale bars: b: 5 mm, c: 50 μm, d: 100 μm, e: 1 mm [147]; (**B**) thermal-inkjet 3D cell printing techniques can mimic the in vivo RGC distribution on retina. (**a**) estimated RGC distribution on retina, including higher cell density near the optic nerve head and lower density at the fovea (*); (**b**) RGC distribution results by inkjet 3D printing; scale bar: 550 μm [148]; (**C**) apparatus applying adjustable hydrostatic pressure to primary RGCs based on Pascal’s law. (**a**) diagram of the apparatus, a transparent reservoir connecting with multiple PDMS chambers, which are containing primary RGC cultures; (**b**,**c**) representative fluorescence images of primary RGCs cultured inside a PDMS chamber at day 3 in vitro. RGCs were positively stained with TUJ1 (green) and BRN3A (Red), which are neuronal-specific and RGC-specific, respectively [149]. Figure republished with permission from each indicated reference ([147] for part A, [148] for part B, [149] for part C).

**Figure 6 micromachines-11-00612-f006:**
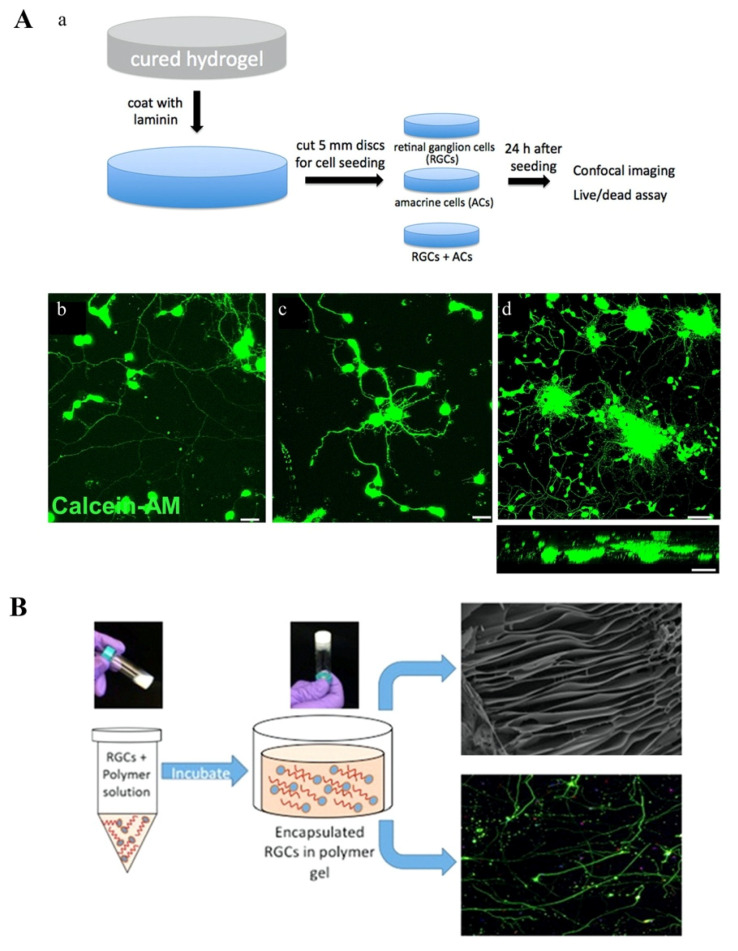
3D models of retina ganglion cell. (**A**): (**a**): Schematic showing the hydrogel preparation procedures and experimental plan for cell seeding and analysis for the tunable hydrogel. RGCs; (**b**) and amacrine cells; (**c**) cultured in tunable hydrogels composed of poly(ethylene glycol) and poly(l-lysine); (**d**) showed both cell types under lower magnification. Both RGCs and amacrine cells migrated into hydrogels and extended neurites in three dimensions. Cells were stained using calcein-AM (green). Scale bar: i, ii: 50 μm, iii: 200 μm [153]; (**B**) injectable hydrogel for RGC regeneration. The mix of RGCs and polymer is at solution state at room temperature. After injected, it would become hydrogel at 37 °C and fix on retina. SEM showed a laminar sheet-like structure of the hydrogel [154]. Figure republished with permission from each indicated reference ([153] for part A, [154] for part B).

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
