# Peer review of "Tissue-Engineered Models for Glaucoma Research"

_micromachines, 2020, doi:10.3390/mi11060612_

Round 1
Reviewer 1 Report
This manuscript by Lu et al. reviewed the current progress in tissue-engineered models for glaucoma research, including models for studying trabecular meshwork, Schlemm’s canal and retinal ganglion cells (RGCs) in 2D and 3D formats with different scaffolds. The authors started with a logical order reviewing the literature from the normal eye physiology to glaucoma research. This is a complete and thoughtful review. I only found some minor errors, and it is suitable for publication in Micromachines after minor revision.
Minor point:
For reference 72: Please change “Crawford Downs, J” to “Downs, J.C”. As you see reference 85, it is “Downs, J.C.” and it is the same author.
- Crawford Downs, J.; Roberts, M.D.; Sigal, I.A. Glaucomatous cupping of the lamina cribrosa: a review of 1177 the evidence for active progressive remodeling as a mechanism. Experimental eye research 2011, 93, 133-140, 1178 doi:10.1016/j.exer.2010.08.004.
- Pant, A.D.; Amini, R. Iris Biomechanics. In Biomechanics of the Eye, Roberts, C.J., Dupps, W.J., Downs, J.C., 1210 Eds. Kugler Publications: Amsterdam, 2018; p. 282.

Author Response
Dear Editor:
Thank you for your feedback on our manuscript (ID: micromachines-842394) entitled, “Tissue-Engineered Models for Glaucoma Research,” which was submitted for consideration as a review article in Micromachines. We have prepared a revised manuscript in which we address the concerns raised by you and the reviewers. Below we provide a point-by-point reply to the comments of the reviewers and a description of how we have incorporated their feedback into our manuscript.
REVIEWER #1
General Comments
This manuscript by Lu et al. reviewed the current progress in tissue-engineered models for glaucoma research, including models for studying trabecular meshwork, Schlemm’s canal, and retinal ganglion cells (RGCs) in 2D and 3D formats with different scaffolds. The authors started with a logical order reviewing the literature from the normal eye physiology to glaucoma research. This is a complete and thoughtful review. I only found some minor errors, and it is suitable for publication in Micromachines after minor revision.
Response:
We thank the reviewer for these insightful comments and suggestions. We have made the necessary revisions to our citations.
Specific comment:
For reference 72: Please change “Crawford Downs, J” to “Downs, J.C”. As you see reference 85, it is “Downs, J.C.” and it is the same author.
- Crawford Downs, J.; Roberts, M.D.; Sigal, I.A. Glaucomatous cupping of the lamina cribrosa: a review of 1177 the evidence for active progressive remodeling as a mechanism. Experimental eye research 2011, 93, 133-140, 1178 doi:10.1016/j.exer.2010.08.004.
- Pant, A.D.; Amini, R. Iris Biomechanics. In Biomechanics of the Eye, Roberts, C.J., Dupps, W.J., Downs, J.C., 1210 Eds. Kugler Publications: Amsterdam, 2018; p. 282.
Response:
We thank the reviewer for this very attentive point, and we apologize for the citation mistake. We have revised as requested (line 295) and updated our EndNote citation file.
REVIEWER #2:
General Comments:
This manuscript reviews the recent advances in tissue-engineered models for glaucoma research. The authors describe the basic physiology of the eye, especially on aqueous humor (AH) outflow and retinal ganglion cells (RGCs), as disruption of AH outflow and degeneration of RGCs contribute to glaucoma progression. The authors then discuss the development of tissue-engineered models of trabecular meshwork (TM) and RGCs for studying mechanisms of resistance to AH outflow and degeneration of RGCs, respectively, for understanding pathogenesis of glaucoma. The challenges and future perspectives associated with the development of tissue-engineered models for glaucoma research are briefly discussed. Overall, the manuscript is interesting and well-written. However, there are a few comments that the authors need to address.
Response:
We thank the reviewer for these insightful comments and suggestions. We have made the necessary revisions and reply to the specific comments point-by-point below.
Specific comments:
- Line 587-588: “In 2015, Dautriche et al. used this model to demonstrate that exposure to TGF-β2 also induces SC cells to undergo an endothelial to mesenchyme transition very similar to that observed in the TM cells of the glaucomatous JCT.” Please discuss on how endothelial to mesenchyme transition of TM and SC cells contribute to glaucoma progression.
Response:
We thank the reviewer for this insightful suggestion. The endothelial to the mesenchyme-like transition of trabecular meshwork (TM) cells was discussed in the section “4.2.1 Excessive Deposition of ECM in the Pre-Glaucomatous Eye” (line 394-400). Briefly, TM cells that have undergone this transition deposit excessive extracellular matrix in the juxtacanalicular tissue (JCT), increasing the hydraulic resistance to trabecular outflow.
With regard to Schlemm’s canal (SC), Dautriche et al.’s showed that TGF-β2 could induce an endothelial to mesenchyme transition in cultured SC endothelial cells. However, to our knowledge, there are no additional in vitro studies of this phenomenon. Furthermore, there is no in vivo evidence currently available to support whether or not an endothelial to mesenchyme transition occurs in the glaucomatous SC. We have clarified this point in lines 391-393.
- Lines 643-644: “When treated with dexamethasone, this model exhibited time-dependent enhancement of trabecular resistance.” How do researchers measure the degree of trabecular resistance in TM cells?
Response:
We thank the reviewer for this comment. Briefly, “trabecular resistance” refers to the hydraulic resistance across an in vivo or in vitro model of the trabecular meshwork (TM). In Waduthanthri et al. (2019), the authors built a perfusion system to circulate media through their 3D model of trabecular physiology at a constant flow rate. They measured the pressure differential across the TM to calculate the hydraulic resistance, or “trabecular resistance,” of the model (reference 116). We recognize that the term “trabecular resistance” might cause some confusion, and we have clarified how we operationalize the term in our revision (lines 691-698).
- It would be great if the authors can provide their opinion on the possibility of using tissue-engineered models in developing drug- and cell-based therapies for glaucoma.
Response:
Again, we are grateful for this insightful suggestion. To address this, we added one paragraph in the conclusion discussing some limitations of current models and potential future applications of these models in drug- and cell-based therapies (lines 1101-1116).
Reviewer 2 Report
This manuscript reviews the recent advances in tissue-engineered models for glaucoma research. The authors describes the basic physiology of eye, especially on aqueous humor (AH) outflow and retinal ganglion cells (RGCs), as disruption of AH outflow and degeneration of RGCs contribute to glaucoma progression. The authors then discusses development of tissue-engineered models of trabecular meshwork (TM) and RGCs for studying mechanisms of resistance to AH outflow and degeneration of RGCs, respectively, for understanding pathogenesis of glaucoma. The challenges and future perspectives associated with the development of tissue-engineered models for glaucoma research are briefly discussed. Overall, the manuscript is interesting and well-written. However, there are a few comments that the authors need to address.
1. Line 587-588: “In 2015, Dautriche et al. used this model to demonstrate that exposure to TGF-β2 also induces SC cells to undergo an endothelial to mesenchyme transition very similar to that observed in the TM cells of the glaucomatous JCT.” Please discuss on how endothelial to mesenchyme transition of TM and SC cells contribute to glaucoma progression.
2. Lines 643-644: “When treated with dexamethasone, this model exhibited time-dependent enhancement of trabecular resistance.” How do researchers measure degree of trabecular resistance in TM cells?
3. It would be great if the authors can provide their opinion on possibility of using tissue-engineered models in developing drug- and cell-based therapies for glaucoma.
Author Response

(The authors gave the same response as above.)
